# Increasing Influenza Vaccination Rates among Patients with Type 2 Diabetes Mellitus in Chongqing, China: A Cross-Sectional Analysis Using Behavioral and Social Drivers Tools

**DOI:** 10.3390/vaccines12080898

**Published:** 2024-08-08

**Authors:** Zhourong Li, Luzhao Feng, Jiang Long, Yu Xiong, Tingting Li, Binshan Jiang, Shuang Yang, Lin Fu, Zumin Shi, Yong Zhao, Li Qi

**Affiliations:** 1Chongqing Municipal Center for Disease Control and Prevention, Chongqing 400799, China; 2021120781@stu.cqmu.edu.cn (Z.L.); 68803648@163.com (J.L.); xiongyu1224@126.com (Y.X.); lee712ting@126.com (T.L.); 2School of Public Health, Chongqing Medical University, Chongqing 400016, China; 2022120852@stu.cqmu.edu.cn (S.Y.); 2023121648@stu.cqmu.edu.cn (L.F.); zhaoyong@cqmu.edu.cn (Y.Z.); 3School of Population Medicine and Public Health, Chinese Academy of Medical Sciences & Peking Union Medical College, Beijing 100730, China; fengluzhao@cams.cn (L.F.); binshan_jiang@student.pumc.edu.cn (B.J.); 4Chongqing Municipal Key Laboratory for High Pathogenic Microbes, Chongqing 400799, China; 5Department of Human Nutrition, College of Health Sciences, QU Health, Qatar University, Doha 2713, Qatar; zumin.shi@gmail.com; 6Infectious Disease Control and Prevention, Chongqing Municipal Center for Disease Control and Prevention, Chongqing 400799, China

**Keywords:** influenza vaccination willingness, type 2 diabetes mellitus patients, behavioral and social drivers

## Abstract

Background: Influenza vaccination is essential for type 2 diabetes mellitus (T2DM) patients due to their higher risks of severe complications and mortality from influenza. This study investigated the willingness of T2DM patients in Chongqing, China, to receive the influenza vaccination during the 2023/2024 season, using behavioral and social drivers (BeSD) tools to improve vaccination uptake in this high-risk group. Methods: A multi-stage sampling method was used to select participants, and face-to-face surveys were conducted in community health centers between March 1 and May 1, 2023. Binary logistic regression was used to analyze the factors influencing vaccination willingness, and standardized scores identified barriers and drivers. Results: Among 1672 T2DM patients, 11.7% had been vaccinated during the 2022/2023 season, and 59.6% were willing to receive the vaccination in the 2023/2024 season. Higher willingness was associated with ethnic minorities (odds ratio [OR], 3.18, 95% confidence interval [CI]: 1.58–6.39), being unemployed individuals (OR 2.69, 95% CI: 1.60–4.52), higher monthly household income per capita (OR 2.72, 95% CI: 1.65–4.50), having diabetes complications (OR 1.76, 95% CI: 1.23–2.51), sufficient vaccine knowledge (OR 1.87, 95% CI: 1.48–2.37), and previous vaccination (OR 7.75, 95% CI: 4.83–12.44). Concerning BeSDs, fear of infecting friends or family members and trust in vaccine efficacy were the predominant drivers, while high vaccine costs were the greatest barrier. Conclusions: Future strategies should focus on improving vaccine knowledge, supporting healthcare workers and peer recommendations, and enhancing vaccination policies. Key interventions such as health education among high-risk groups, such as unemployed individuals, advocacy campaigns, pay-it-forward strategies, and policies for free vaccination could improve coverage in Chongqing.

## 1. Background

Patients with type 2 diabetes mellitus (T2DM) are generally more susceptible to influenza and have an elevated risk of developing severe complications, including pneumonia, premature death, acute cardiovascular complications, and hospitalizations [1,2,3]. Influenza vaccination has been strongly recommended as a primary imperative for diabetes patients, serving as the principal means of reducing or counteracting influenza mortality and morbidity in the community [4,5]. A meta-analysis of observational studies revealed that adults and older individuals with diabetes who received the influenza vaccination had a lower risk of hospitalization for pneumonia [6]. A cohort study conducted in Norway found that hospitalization and mortality rates for pandemic influenza were 78% and 25% lower, respectively, in vaccinated patients than in non-vaccinated patients with T2DM [2]. However, the influenza vaccination rate has consistently remained low in China, at only 4.0%, and was even lower during the COVID-19 pandemic [7]. Specifically, influenza vaccination coverage in China was 3.16% in the 2020–2021 and 2.47% in the 2021–2022 epidemic seasons [8].

Vaccine hesitancy, which refers to delayed acceptance or refusal of vaccination despite the availability of vaccination services, has been recognized as one of the leading 10 global health concerns [9]. Factors related to influenza vaccine (IV) uptake in risk-group populations have previously been investigated, mainly based on the knowledge, attitude, belief, and practice of health education research [10,11,12,13]. However, the practical issues and social factors affecting vaccine uptake have not been comprehensively explored. In terms of vaccination surveys, the behavioral and social drivers (BeSDs) released by the World Health Organization (WHO) in May 2022 are vital tools that help ensure that there is a focus not only on what people think and feel but also that sufficient attention is given to social influences and practical issues that affect vaccine uptake [14]. The BeSD framework has been widely utilized in the surveys assessing the intention to receive the COVID-19 vaccination as well as the willingness of healthcare workers to receive IVs [15,16].

Chongqing is a large municipality with a heavy disease burden from seasonal influenza and diabetes [17,18]. Therefore, the significance of influenza vaccination in patients with T2DM cannot be overstated. A systematic review showed that limited vaccine knowledge and negative attitudes toward healthcare services may impede influenza vaccination, whereas healthcare trust promotes vaccine uptake [19]. However, the influenza vaccination rate and its influencing factors in patients with T2DM in Chongqing are not well known. The aim of this study was thus to explore the willingness and determinants of patients with T2DM to be vaccinated against influenza in Chongqing based on the BeSD framework, to improve influenza vaccination coverage, and to promote the health management of patients with T2DM in the community.

## 2. Methods

### 2.1. Study Design and Sample

This cross-sectional study was conducted between 1 March and 1 May 2023, in Chongqing, China. A multi-stage sampling method was used to select participants. Randomization was performed using a computer-generated table of random numbers to select five from 39 districts/counties, and one urban and one rural community health center were randomly selected as survey sites from each district/county. The convenience sampling method was used for the face-to-face surveys. The inclusion criteria for patients with T2DM were as follows: (1) diagnosed with T2DM by practitioners, (2) local residence for at least 6 months, (3) age over 18 years, and (4) informed consent and cooperation in completing the questionnaire. The exclusion criteria were individuals with intellectual disabilities, psychiatric disorders, or hearing and speech impairments who were incapable of independently participating in the survey.

According to the Standards for National Basic Public Health Services in China, people diagnosed with T2DM are provided with free-of-charge regular follow-up care and fasting plasma glucose monitoring in primary care facilities [20]. In this study, a total of 10 community health centers were selected from five districts/counties in Chongqing. A flowchart of the sampling procedure is shown in Figure 1. Two trained general practitioners, designated as investigators at each site, conducted face-to-face surveys and reviewed and recorded the answers on Questionnaire Star (Changsha Ranxing Information Technology Co., Ltd., Changsha, China), a widely used online survey platform in China (Appendix A). Each survey lasted for approximately 8–12 min.

The required sample size was estimated using the following sample size calculation formula:
$\;n=\left(\frac{{Z_{\propto /2}}^{2}}{d^{2}}\right)*p*\left(1-p\right)$. Referring to the influenza vaccination rate of patients with chronic diseases in China,
p was 0.094, the margin of error
d = 0.20
p, and
$Z_{\propto /2}$ = 1.96 [21]. The sample size was calculated at 926. Considering sampling errors and invalid responses, an additional 20% was added to the estimated sample size, resulting in a final target sample of 1112 individuals.

In this study, a total of 1841 individuals with T2DM were invited to participate. Participants who failed the attention check or did not complete all the questions were excluded. After excluding invalid questionnaires, 1672 participants were included in the final analysis, with a response rate of 90.8%.

### 2.2. Data Collection

The questionnaire was designed according to the BeSD framework, the technical guidelines for seasonal influenza vaccination in China (2022–2023), and research on the field of influenza vaccination [5,14,16,22]. The final version was also developed based on insights from experts in epidemiology and health behavior. The questionnaire consisted of four sections, as follows:(1)Demographic characteristics (12 questions), including age, sex, ethnicity, residence, education, occupation, marital status, average monthly household income per capita (which refers to the total monthly income of a household divided by the number of household members), height and weight, duration of diabetes, diabetes complications, and other chronic diseases. Body mass index (BMI) was calculated by dividing body weight in kilograms by the square of body height in meters. In this study, we defined adults with a BMI of
<18.5 kg/m^2^ as underweight, those with a BMI between
≥24.0 and
<28.0 kg/m^2^ as overweight, and those with a BMI of
≥28.0 kg/m^2^ as obese [23].(2)Influenza history and vaccination status in the past year (5 questions), including the history of influenza and vaccination records. For those who received influenza vaccinations during the past year, additional information was sought regarding the vaccination site, adverse effects, and vaccination payment.(3)Practice and knowledge of influenza and influenza vaccination (13 questions), encompassing the following aspects: (a) learning about influenza prevention and control, (b) acquiring knowledge about influenza vaccination, (c) recognizing influenza vaccination as the most effective way to prevent influenza, and (d) identifying priority groups for influenza vaccination (10 questions). For the knowledge of influenza vaccination (the final two questions), one point was awarded for each correct response, with incorrect responses receiving no points. The total score for the questions ranged from 0 to 11. Attaining a score of seven or above (60%) was considered to indicate good IV knowledge, whereas scores below 7 were considered to indicate poor knowledge. Higher scores indicated better knowledge.(4)To explore the determinants of IV uptake in patients with T2DM, we further investigated the willingness to receive influenza vaccination and its influencing factors (15/18 questions), which included the willingness to be vaccinated in the 2023/2024 influenza season, drivers and barriers to receiving seasonal influenza vaccination (in relation to their thinking and feeling, social processes, and practical issues), and willingness to receive influenza vaccination when a free vaccination policy was implemented. The reasons for willingness (11 questions) or hesitancy (14 questions) to receive seasonal influenza vaccination were evaluated on a 3-point scale, with answers ranging from 0 = “disagree” to 2 = “agree”, to identify factors that affected influenza vaccination. The higher the score, the greater the impact of the factor. The average score for each question was calculated by dividing the total score by the number of participants; the standardized score for each dimension was the total dimension score divided by the number of questions.

The primary outcome was the willingness to be vaccinated in the 2023/2024 influenza season, with options being “definitely yes”, “not sure”, and “definitely not.” According to the definition of vaccine hesitancy, individuals who selected either of the latter two options were required to complete the hesitancy scale, while those choosing “definitely yes” were required to respond to the acceptance scale.

In total, 165 questionnaires were collected based on a pilot survey conducted in February 2023. The Cronbach’s α coefficient and Kaiser-Meyer-Olkin (KMO) test values for the acceptance scale were 0.944 and 0.891, while the Cronbach’s α and KMO test values were 0.919 and 0.817 for the hesitancy scale, respectively, which indicate a satisfactory level of reliability and validity.

### 2.3. Ethics Approval and Consent to Participate

Oral consent was obtained from all participants at the beginning of the investigation, and this study was approved by the Ethics Committee of Chongqing Municipal Center for Disease Control and Prevention (approval number: CQCDCLS (2021) 026) and Chongqing Medical University (approval number: 2023084).

### 2.4. Statistical Analyses

Microsoft Excel (version 2019; Redmond, WA, USA) was used to collect the data. All analyses were conducted using STATA/MP software (version 17.0; College Station, TX, USA). Descriptive statistics, such as frequency and percentages, were calculated for categorical variables to show the distribution of demographic characteristics, and the mean (standard deviation (SD)) was used for quantitative variables. The analysis of variance and the chi-square test were used to assess the differences. Binary logistic regression analysis was employed to identify determinants of willingness to receive influenza vaccination among the participants with T2DM. Variables that were statistically significant in the univariate analysis (*p* < 0.05) were subsequently incorporated into the multivariate logistic regression model to adjust for potential confounders and ascertain independent predictors. Percentage bar charts were used to show the distribution of barriers to and drivers of vaccination. Standardized scores were used to evaluate the weights of the domains affecting influenza vaccination. Odds ratios (ORs) with corresponding 95% confidence intervals (CIs) were calculated to identify the determinants. Statistical significance was set at a *p*-value < 0.05 (two-tailed).

## 3. Results

### 3.1. Demographic Characteristics and Influenza History of the Participants with T2DM

The demographic characteristics of the participants are presented in Table 1. A total of 1672 participants with T2DM were included in this study. The mean age was 65.96 ± 10.04 years. The mean duration of T2DM was 6.99 ± 5.53 years. Among participants, 1051 (62.9%) were female, 931 (55.7%) lived in urban areas, and 873 (52.2%) were workers or farmers. A total of 969 (58.0%) participants had less than primary school education, and 892 (53.4%) had an average monthly household income per capita below 2000 RMB. A total of 192 (11.5%) participants had diabetes complications, and 887 (53.1%) had other chronic diseases.

In the 2022/2023 influenza season, 688 (41.1%) participants had influenza, and only 196 (11.7%) reported having been vaccinated against influenza. Of the 196 vaccinated participants, 121 (61.7%) were vaccinated at community health centers, 145 (74.0%) were vaccinated at their own expense, and 43 (21.9%) received medical reimbursement (Table 2). Among the 1672 respondents, 989 (59.2%) intended to receive an influenza vaccination during the 2023/2024 influenza season.

### 3.2. Knowledge and Practice of Influenza Vaccination among the Participants with T2DM

Among the 1672 participants, the score for influenza vaccination knowledge averaged 6.65 ± 3.08, with 55.5% having satisfactory knowledge. Participants who were willing to receive influenza vaccination scored higher than those in the hesitancy group (average score: 7.37 vs. 5.62, respectively), demonstrating a statistically significant difference (*p* < 0.001) (Table 1). Of the participants, 52.8% considered that influenza vaccination was the most effective way to prevent influenza. Healthcare workers were identified as the main priority group for influenza vaccination, with the highest proportion (78.5%), followed by adults aged 60 years and above (77.5%), and those with diabetes (75.0%), while the lowest proportion was for pregnant women (47.1%) (Table 3).

Additionally, 20.6% of the participants actively learned about the prevention and control of influenza, whereas the proactive acquisition of knowledge about influenza vaccination was slightly lower (18.7%) (Table 3).

### 3.3. Factors Associated with the Willingness to Receive Influenza Vaccination

Table 4 shows the factors influencing the willingness of the participants with T2DM to receive influenza vaccination during the 2023/2024 influenza season. Notably, participants of minority ethnicities (OR, 3.18, 95% CI: 1.58–6.39), unemployed individuals (OR, 2.69, 95% CI: 1.60–4.52), individuals with an average monthly household income per capita exceeding 5000 RMB (OR, 2.72, 95% CI: 1.65–4.50), those with diabetes complications (OR, 1.76, 95% CI: 1.23–2.51), those with sufficient knowledge of influenza vaccination (OR, 1.87, 95% CI: 1.48–2.37), and those who had been vaccinated before (OR, 7.75, 95% CI: 4.83–12.44) were more inclined to receive influenza vaccination. However, individuals residing in urban areas (OR, 0.68, 95% CI: 0.53–0.89) and those with T2DM for more than 7 years (OR, 0.74, 95% CI: 0.59–0.93) were hesitant regarding influenza vaccination.

### 3.4. Main Drivers and Barriers for Willingness to Receive Influenza Vaccination Based on the BeSD Framework

Figure 2 and Figure 3 present the drivers of and barriers to IV uptake within the BeSD framework, respectively. Of the 989 participants who were willing to receive influenza vaccination in the 2023/2024 influenza season, thinking and feeling exerted the greatest impact on their inclination to receive vaccination, yielding a standardized score of 1.44
± 0.54 (Table 5).

Among the 11 driving factors, fear of infecting friends or family members and trust in the effectiveness of influenza vaccination were the two primary drivers, both with a mean score of 1.50. Doctors’ recommendations and confidence in the safety of the IV were the second most influential factors (1.43 vs. 1.43, respectively) (Figure 2).

Among those in the hesitancy group, social processes emerged as the most critical factor, with an average score of 0.97
 ± 0.67 (Table 5). The high cost of the IV was the main reason (1.29) for hesitancy to receive influenza vaccination (Figure 3), followed by a lack of recommendations by friends or family members (1.08), uncertainty about the timing of influenza vaccination (1.08), and belief that influenza would not cause severe illness (1.04).

## 4. Discussion

Patients with T2DM are prioritized for influenza vaccination, both in China and abroad. However, in our study, the vaccination rate was only 11.7% in the 2022/2023 influenza season, far lower than that in high-income countries, such as the Netherlands (74.8%) [24], Spain (65.7%) [25], and Korea (59.6%) [26], and in relation to the target set by the WHO of 75% [27,28]. In this study, 59.2% of those with T2DM were willing to be vaccinated against influenza in the 2023/2024 influenza season, which was much higher than in other cities such as Shenzhen [29], Ningbo [30], and Changsha [31] in previous seasons. The heightened focus on respiratory infectious diseases due to the COVID-19 pandemic is a plausible explanation. Regarding the drivers and barriers to influenza vaccination in relation to the BeSD framework, fear of infecting friends or family members and trust in the effectiveness of the IV were identified as the predominant motivating factors, while the high cost of the IV was identified as the greatest barrier. Consequently, targeted interventions are imperative to address the discrepancies in vaccination willingness and behavior among those with T2DM.

Our study revealed that approximately half of the participants with T2DM had satisfactory knowledge of influenza vaccination, with most respondents regarding healthcare workers as the highest priority group, while more than half did not consider pregnant women as a priority group. Vaccination protection among healthcare workers is crucial for mitigating the intensity and spread of infection and maintaining the robustness of the healthcare system, thus effectively reducing absenteeism rates and the workdays lost among these workers [32,33]. However, pregnant women were not well recognized as a group needing attention, probably because most of the respondents were older adults who tended to be more aware of the needs of their own population group. Although people with diabetes are ranked among the leading three priority groups for vaccination, a significant gap remains in translating this prioritization into actual vaccination behavior.

Notably, ethnicity, occupation, income, diabetes complications, influenza vaccination knowledge, history of influenza vaccination, residence, and duration of diabetes were significant factors affecting vaccination. We found that the participants with T2DM and with good knowledge of influenza vaccination were more likely to be vaccinated than those with poor knowledge, which is consistent with the findings of Olatunbosun et al. [34] and Ibraheem et al. [35]. Consequently, sufficient knowledge of influenza vaccination appears to have contributed to a high seasonal influenza vaccination uptake. Diabetes complications were a protective factor in terms of influenza vaccination willingness, in contrast to the findings of a previous study conducted in Saudi Arabia [35]. A possible explanation is that the participants with diabetes complications were likely to be more aware of the risks associated with influenza infection.

This study also revealed that rural residents exhibited a greater inclination to receive influenza vaccination, similar to the findings of a previous study conducted in Chongqing [36]. The reason for this may be the difference in attitudes toward influenza vaccination between urban and rural residents. Urban residents may perceive themselves as possessing greater health literacy and enjoying easier access to abundant medical resources, potentially resulting in underestimating the severity of influenza. Conversely, rural residents may demonstrate heightened concerns about the financial implications of falling ill, thereby exhibiting a greater inclination to receive the influenza vaccination. Vaccination decisions are frequently anchored in past experiences with influenza vaccination [25,37,38]. Our findings concurred with these results, indicating that a previous history of influenza vaccination had the highest OR value (7.62) among all influencing factors. This suggested that those participants with T2DM and a history of influenza vaccination may have an increased perception of the benefits associated with vaccination. Moreover, the systematic review and meta-analysis suggested that vaccination in the previous year attenuates vaccine effectiveness, but vaccination in two consecutive years provides superior protection compared to no vaccination [39]. Therefore, it is essential to develop targeted strategies to promote annual influenza vaccination uptake among those with T2DM.

Based on the BeSD framework, the intention to receive influenza vaccination was determined by thinking and feeling, social processes, and practical issues. Our findings, which confirm previous research findings, underscore the psychological determinants (thinking and feeling) that appear to be the most significant factors in vaccine acceptance [40,41]. Trust in the effectiveness of influenza vaccination was one of the key drivers identified in this study. A systematic review showed that seasonal influenza vaccination reduced the risk of hospitalization and mortality in patients with diabetes, particularly those aged 65 years and older [42]. Therefore, public health education should prioritize highlighting the safety and effectiveness of influenza vaccination and emphasizing the indirect protective benefits of the IV for susceptible community members. Our study also found that social processes played a key role in influenza vaccination intentions. Recommendations from healthcare workers contribute to increasing influenza vaccination coverage and have been well documented [43,44]. Despite concerns about adverse reactions to the vaccine in their patients, most healthcare workers report impediments in terms of recommending influenza vaccination [16]. Healthcare workers who had been vaccinated during the past year demonstrated a greater propensity to recommend influenza vaccination to children and older adults [45]. Hence, ensuring that healthcare workers are annually vaccinated and equipped with information for which there is professional consensus, scientifically based knowledge, and sound perspectives through comprehensive training is imperative for influenza prevention and the promotion of vaccination against influenza in prioritized groups. Recommendations from friends or family members were identified as primary motivators. Future research should investigate the positive influence of peer recommendations on the intention to vaccinate against influenza.

The high cost of the IV served as the principal impediment to vaccine uptake in our investigation, while providing free influenza vaccination proved to be a compelling incentive, leading to a significant boost in the willingness rate, with an approximate increase of 25.0%. Our study thus emphasizes the importance of addressing cost-related obstacles to enhancing vaccine uptake. Watkinson et al. [46] concluded that income deprivation is associated with lower influenza vaccination uptake. Further, an investigation conducted in Hong Kong found that free or subsidized influenza vaccinations were one of the most frequently reported reasons for influenza vaccination [47]. A quasi-experimental trial undertaken by Wu et al. in China revealed that the pay-it-forward intervention (a free IV and an opportunity to donate financially to support the vaccination of other individuals) seemed to be effective in improving influenza vaccination uptake and community engagement, substantially enhancing participant confidence in the importance, safety, and effectiveness of the vaccine [48]. This finding suggests the potential of establishing an urban-to-rural subsidization mechanism to support influenza vaccination in poor areas. According to the National Bureau of Statistics of China, in 2023, the average annual income in urban areas was approximately 51,821 RMB, while in rural areas, it was around 21,691 RMB [49]. This significant disparity highlights the varying financial burdens that vaccination costs may impose on different populations. Consequently, pay-it-forward and free vaccination policies should be implemented among those with T2DM to reduce vaccination inconsistencies caused by socioeconomic disparities. Vaccine co-administration has been found to be an effective way to improve vaccination coverage as well [50]. A retrospective observational study showed that the administration of combined pneumococcal and influenza vaccination appeared safe and significantly reduced the risk of mortality [51]. Future studies should focus on the potential impact of cost incentives in combining immunization regimens to enhance intravenous uptake.

Our study is the first to use the BeSD framework to survey influenza vaccination willingness among those with T2DM and identify its influencing factors. However, this study has some limitations. First, our survey participants were recruited from a community health management program, leading to constraints in terms of geographical representation; therefore, our findings cannot be generalized to all individuals with T2DM in China. Second, although the investigators repeatedly checked participant answers, recall bias is inevitable in self-reported surveys. Third, the capacity to draw direct causal inferences was constrained by the use of cross-sectional survey data. Future research should consider conducting multicenter studies to ensure broader geographic representation, implement more stringent quality control measures, and apply longitudinal or experimental research designs to evaluate the effects of various interventions on influenza vaccination.

## 5. Conclusions

In summary, our study provides insights into the drivers and barriers influencing the willingness to receive influenza vaccination among individuals with T2DM in Chongqing, China. Key factors included minority ethnicities, unemployment, income, diabetes complications, vaccination knowledge, and previous vaccination history. The BeSD survey further highlighted the importance of perceived risks, vaccine efficacy beliefs, and particularly cost concerns as significant barriers. To improve vaccination coverage, future strategies should prioritize health education on vaccine safety and effectiveness, focus on high-risk groups, advocate for healthcare professionals and peer recommendations, and address cost barriers. Implementing medical insurance coverage or free vaccination policies for T2DM patients is also essential. Additionally, future research should explore the impact of financial incentives for combined vaccination programs and pay-it-forward strategies to enhance vaccine uptake.

## Figures and Tables

**Figure 1 vaccines-12-00898-f001:**
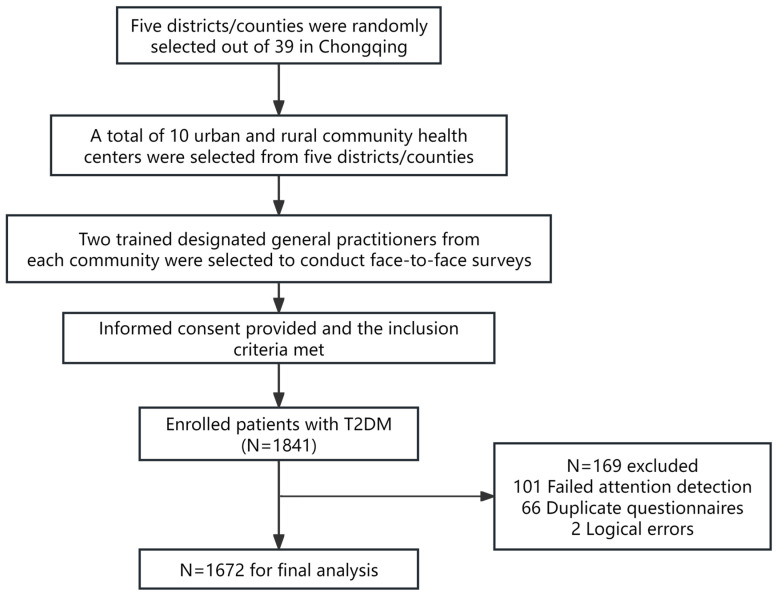
Sampling flowchart of participants in Chongqing, China.

**Figure 2 vaccines-12-00898-f002:**
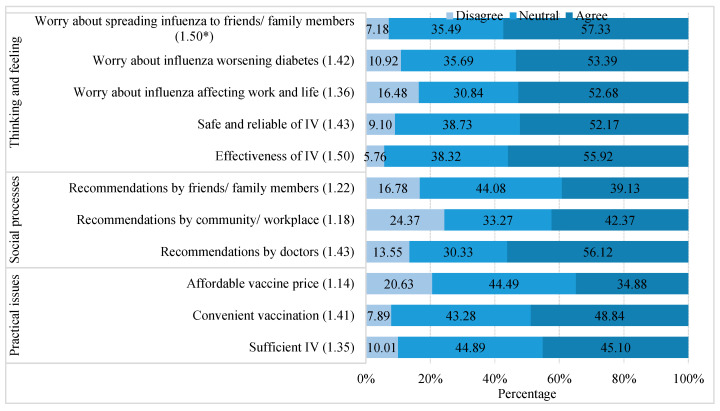
Drivers of influenza vaccination among the participants with T2DM in Chongqing (n = 989). Note: * The average score of each item; T2DM, type 2 diabetes mellitus; IV, influenza vaccine.

**Figure 3 vaccines-12-00898-f003:**
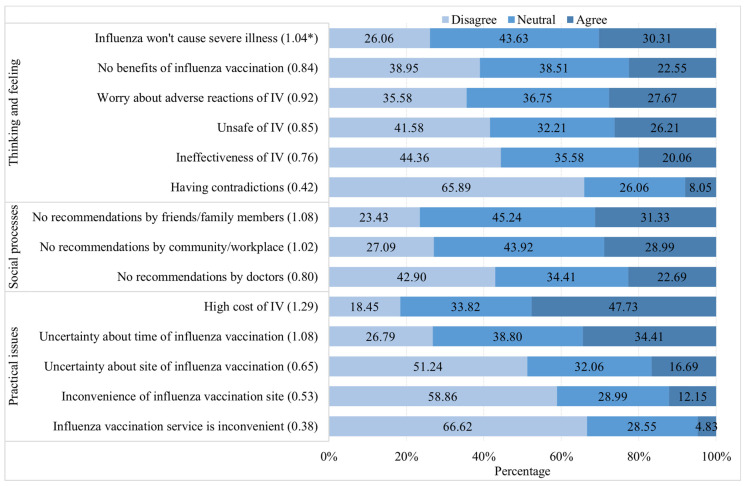
Barriers to influenza vaccination among the participants with T2DM in Chongqing (n = 683). Note: * The average score of each item; T2DM, type 2 diabetes mellitus; IV, influenza vaccine.

**Table 1 vaccines-12-00898-t001:** Respondent characteristics and the willingness to be vaccinated in Chongqing, China, 2023, n (%) (n = 1672).

Characteristics	Total	Willingness to Be Vaccinated		*p* ^1^
		Acceptance	Hesitancy	
Total	1672 (100.0)	989 (59.2)	683 (40.8)	
Age (mean ± SD)	65.96 (10.04)	65.23 (10.26)	67.02 (9.61)	<0.001
<60 years	461 (27.6)	307 (66.6)	154 (33.4)	<0.001
≥60 years	1211 (72.4)	682 (56.3)	529 (43.7)	
Duration of T2DM (mean ± SD)	6.99 (5.53)	6.60 (5.1)	7.55 (6.0)	<0.001
<7 years	954 (57.1)	598 (62.7)	356 (37.3)	<0.001
≥7 years	718 (42.9)	391 (54.5)	327 (45.5)	
Sex				0.78
Male	621 (37.1)	370 (59.6)	251 (40.4)	
Female	1051 (62.9)	619 (58.9)	432 (41.1)	
Ethnicity				<0.001
Han	1591 (95.2)	919 (57.8)	672 (42.2)	
Minority ethnicity	81 (4.8)	70 (86.4)	11 (13.6)	
Residence				<0.001
Rural	741 (44.3)	481 (64.9)	260 (35.1)	
Urban	931 (55.7)	508 (54.6)	423 (45.4)	
Marital status				<0.001
Single	11 (0.7)	6 (54.5)	5 (45.5)	
Married	1431 (85.6)	878 (61.4)	553 (38.6)	
Widowed	201 (12.0)	89 (44.3)	112 (55.7)	
Divorced	29 (1.7)	16 (55.2)	13 (44.8)	
Education level				0.36
Primary school and below	969 (58.0)	579 (59.8)	390 (40.2)	
Junior high school	516 (30.9)	297 (57.6)	219 (42.4)	
Senior high school or equivalent	132 (7.9)	75 (56.8)	57 (43.2)	
College/bachelor’s degree or above	55 (3.3)	38 (69.1)	17 (30.9)	
Occupation				<0.001
Office worker	108 (6.5)	62 (57.4)	46 (42.6)	
Businessman	77 (4.6)	36 (46.8)	41 (53.2)	
Worker/farmer	873 (52.2)	488 (55.9)	385 (44.1)	
Retiree	142 (8.5)	51 (35.9)	91 (64.1)	
Unemployed individual	472 (28.2)	352 (74.6)	120 (25.4)	
Body mass index				0.61
Underweight	21 (1.3)	10 (47.6)	11 (52.4)	
Normal	666 (39.8)	387 (58.1)	279 (41.9)	
Overweight	688 (41.2)	413 (60.0)	275 (40.0)	
Obese	297 (17.8)	179 (60.3)	118 (39.7)	
Average monthly household income per capita				<0.001
<2000 RMB	892 (53.4)	522 (58.5)	370 (41.5)	
2000–5000 RMB	638 (38.2)	353 (55.3)	285 (44.7)	
>5000 RMB	142 (8.5)	114 (80.3)	28 (19.7)	
Diabetes complications				0.004
Yes	192 (11.5)	132 (68.8)	60 (31.3)	
No	1480 (88.5)	857 (57.9)	623 (42.1)	
Other chronic diseases				0.050
Yes	887 (53.1)	505 (56.9)	382 (43.1)	
No	785 (46.9)	484 (61.7)	301 (38.3)	
Influenza vaccine knowledge	6.65 (3.08)	7.37 (2.53)	5.62 (3.48)	<0.001
Fail	744 (44.5)	362 (48.7)	382 (51.3)	<0.001
Pass	928 (55.5)	627 (67.6)	301 (32.4)	

^1^ Chi-square test and analysis of variance showing distribution by willingness to be vaccinated across demographic characteristics. Note: n: number, T2DM: type 2 diabetes mellitus, 1 USD ≈ 7.12 RMB (as of the time of this study).

**Table 2 vaccines-12-00898-t002:** Influenza history and vaccination status among the participants with T2DM in the 2022/2023 influenza season, n (%) (n = 1672).

Variables	Total	Willingness to Be Vaccinated		*p* ^1^
		Acceptance	Hesitancy	
Total	1672 (100.0)	989 (59.2)	683 (40.8)	
History of influenza				<0.001
Severe	59 (3.5)	39 (66.1)	20 (33.9)	
Mild	629 (37.6)	406 (64.5)	223 (35.5)	
None	984 (58.8)	544 (55.3)	440 (44.7)	
History of influenza vaccination				<0.001
Yes	196 (11.7)	173 (88.3)	23 (11.7)	
No	1476 (88.3)	816 (55.3)	660 (44.7)	
Vaccination site (n = 196)				0.77
Hospital	70 (35.7)	61 (87.1)	9 (12.9)	
Local CDCs	5 (2.6)	4 (80.0)	1 (20.0)	
Community health centers	121 (61.7)	108 (89.3)	13 (10.7)	
Adverse effects (n = 196)				0.091
Yes	2 (1.0)	1 (50.0)	1 (50.0)	
No	194 (99.0)	172 (88.7)	22 (11.3)	
Payment of influenza vaccination (n = 196)				0.31
Self-paid	145 (74.0)	127 (87.6)	18 (12.4)	
Employer paid	8 (4.1)	6 (75.0)	2 (25.0)	
Medical insurance	43 (21.9)	40 (93.0)	3 (7.0)	

^1^ Chi-square test showing distribution by the willingness to be vaccinated across the history of influenza vaccination. Note: n, number; CDCs, Centers for Disease Control and Prevention; T2DM, type 2 diabetes mellitus.

**Table 3 vaccines-12-00898-t003:** Knowledge and practice of influenza and influenza vaccination among the participants with T2DM, n (%) (n = 1672).

Knowledge and Practice of Influenza and Influenza Vaccination		Yes	No
Practice	Learning about influenza prevention and control	345 (20.6)	1327 (79.3)
	Acquiring knowledge about influenza vaccination	314 (18.7)	1358 (81.2)
Knowledge	Recognizing influenza vaccination as the most effective method for preventing influenza	883 (52.8)	789 (47.2)
	The priority groups for influenza vaccination		
	(1) Healthcare workers	1312 (78.5)	360 (21.5)
	(2) Adults ≥ 60 years of age	1295 (77.5)	377 (22.6)
	(3) Individuals with diabetes	1254 (75.0)	418 (25.0)
	(4) Individuals with chronic respiratory diseases	1245 (74.5)	427 (25.5)
	(5) Individuals with high blood pressure	1220 (73.0)	452 (27.0)
	(6) People living in nursing homes or welfare homes and staff who take care of vulnerable, at-risk individuals	1171 (70.0)	501 (30.0)
	(7) People who work in nursery institutions, primary and secondary schools, and supervision places	1158 (69.3)	514 (30.7)
	(8) Participants and support personnel for large-scale events	1042 (62.3)	630 (37.7)
	(9) Children 6–59 months of age	1011 (60.5)	661 (39.5)
	(10) Pregnant women	788 (47.1)	884 (52.9)

Note: n, number; T2DM, type 2 diabetes mellitus.

**Table 4 vaccines-12-00898-t004:** Binary logistic regression analysis for influential factors associated with the willingness of the participants with T2DM to receive influenza vaccination.

Characteristics		OR (95% CI) *	*p*
Demographic characteristics			
Age			
	<60 years (Ref)	1	
	≥60 years	0.90 (0.69–1.18)	0.442
Ethnicity			
	Han (Ref)	1	
	Minority ethnicity	3.18 (1.58–6.39)	0.001
Residence			
	Rural (Ref)	1	
	Urban	0.68 (0.53–0.89)	0.004
Marital status			
	Single (Ref)	1	
	Married	1.42 (0.40–5.05)	0.585
	Widowed	0.72 (0.20–2.64)	0.622
	Divorced	1.44 (0.32–6.53)	0.637
Occupation			
	Office worker (Ref)	1	
	Businessman	0.99 (0.51–1.93)	0.976
	Worker/farmer	1.17 (0.70–1.95)	0.558
	Retiree	0.59 (0.32–1.07)	0.084
	Unemployed individual	2.69 (1.60–4.52)	0.000
Average monthly household income per capita			
	<2000 RMB (Ref)	1	
	2000–5000 RMB	0.94 (0.73–1.22)	0.642
	>5000 RMB	2.72 (1.65–4.50)	0.000
History of chronic diseases			
Duration of diabetes			
	<7 years (Ref)	1	
	≥7 years	0.74 (0.59–0.93)	0.011
Diabetes complications			
	No (Ref)	1	
	Yes	1.76 (1.23–2.51)	0.002
Other chronic diseases			
	No (Ref)	1	
	Yes	0.92 (0.73–1.15)	0.460
Influenza history and vaccination status			
History of influenza			
	Severe (Ref)	1	
	Mild	0.76 (0.41–1.39)	0.371
	No	0.56 (0.31–1.02)	0.056
History of influenza vaccination			
	No (Ref)	1	
	Yes	7.75 (4.83–12.44)	0.000
Influenza vaccine knowledge			
	Fail (Ref)	1	
	Pass	1.87 (1.48–2.37)	0.000

* Reference: hesitancy group. Note: OR, odds ratio; CI, confidence interval; T2DM, type 2 diabetes mellitus, 1 USD ≈ 7.12 RMB (as of the time of this study).

**Table 5 vaccines-12-00898-t005:** Distribution of behavioral and social drivers of influenza vaccination among the participants with T2DM, n (%).

Domains	Disagree	Neutral	Agree	Standardized Score (Mean ± SD) *
Drivers				
Thinking and feeling	489 (9.9)	1771 (35.8)	2685 (54.3)	1.44 (0.54)
Social processes	541 (18.2)	1065 (35.9)	1361 (45.9)	1.28 (0.64)
Practical issues	381 (12.8)	1312 (44.2)	1274 (42.9)	1.30 (0.54)
Barriers				
Thinking and feeling	1724 (42.1)	1453 (35.5)	921 (22.5)	0.80 (0.56)
Social processes	638 (31.1)	844 (41.2)	567 (27.7)	0.97 (0.67)
Practical issues	1516 (44.4)	1108 (32.4)	791 (23.2)	0.79 (0.49)

* Standardized score: total dimension score divided by the number of questions. Note: n, number; T2DM, type 2 diabetes mellitus.

## Data Availability

The datasets used and/or analyzed during the current study are available from the corresponding author upon reasonable request.

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
