# Peer review of "Increasing Influenza Vaccination Rates among Patients with Type 2 Diabetes Mellitus in Chongqing, China: A Cross-Sectional Analysis Using Behavioral and Social Drivers Tools"

_vaccines, 2024, doi:10.3390/vaccines12080898_

Round 1

Reviewer 1 Report

Comments and Suggestions for Authors

The article is interesting and can provide important information for the development of prevention programs, but the conclusions are not sufficiently clearly expressed and the degree of generality of the conclusions is not clearly presented, starting from the experience of other authors in the general understanding of vaccination adherence, including correlated with treatment adherence in general.

Author Response

Dear reviewer,

We really appreciate your favorable and constructive comments that have helped us to improve this manuscript. Meanwhile, we appreciate having the gracious opportunity to revise the manuscript. After careful consideration, we provided point-by-point responses to your comments. And in the revised version of the manuscript, we marked all changes in red so that you can find them quickly. Additionally, we’ve edited the text for language, grammar, and improved clarity. Ambiguous statements have been rephrased for precision, and all terms and statistical data have been checked for accuracy.

If you think the manuscript still needs more revisions, please continue to give us feedback. Thank you very much. Looking forward to your reply.

                                                                                                                             Best regards,

                                                                                                                             Jul 28th, 2024

Reviewer 1

Comments and Suggestions for Authors

The article is interesting and can provide important information for the development of prevention programs, but the conclusions are not sufficiently clearly expressed and the degree of generality of the conclusions is not clearly presented, starting from the experience of other authors in the general understanding of vaccination adherence, including correlated with treatment adherence in general.

Response: Thank you for your valuable feedback and insightful comments on our manuscript. We appreciate your suggestions for improving the clarity and generality of our conclusions. We’ve revised the conclusions section to enhance clarity and better present the degree of generality of our findings, the revised conclusions in the abstract (Page 1, Line 36-37, marked in red) and the conclusions section of the manuscript (Page 13, Line 388-399, marked in red) .

Reviewer 2 Report

Comments and Suggestions for Authors

The Manuscript „Enhancing influenza vaccination among patients with type 2 diabetes mellitus in Chongqing, China: a cross-sectional analysis using behavioural and social driver tools” represents an attempt to operationalize the Behavior and Social Drivers (BeSD) tools to better understand barriers and motivators to influenca vaccination among patients with type 2 diabetes mellitus and propose methods to increase the vaccination coverage.

I congratulate the Authors on a very interesting study and very good presentation of the results. The topic of the paper is very relevant, the tools used in the research are up-to-date and studies of this type are more than welcome in the field of using behavior science to better understand vaccination behavior and improve it, especially in vulnerable populations. The Authors should address several comments to better explain their work, and some minor corrections, after which the paper would be recommended for publication.

Abstract

The abstract is well written and includes all the primary information necessary to understand the work. Two major comments arise here:

1)     The conclusions seem like they do not follow the Results section. If the main drivers of vaccination (modifiable) were: cost/income, complications, and previous vaccination – I suggest the Authors focus on those aspects in the conclusions. Vaccine efficacy and other social drivers are not mentioned with OR and CI, which is a pity – that is useful information. Being a minority, household and unemployed could also be mentioned as social determinants of health

2)     Please explain or rewrite: “household individual”

3)     Methods: if possible, it would be great to understand how vaccination behavior was measured (scale 1-5 for willingness) and how was the BeSD questionnaire constructed (e.g. used as is in the WHO document)

Introduction

Line 42 – not sure all the complications are “post-infection”, they are just “complications of infection”

Please end the Introduction with a clear AIM of the study. E.g. The aim of this study was to identify…

Methods

Please provide an English version of the full questionnaire as Supplementary material to allow for reproducibility and repeatability of your results.

Please provide an explanation why you opted for a 3 point (Likert) scale for the willingness or hesitancy scale and how you decided for the outcome options (definitely yes, not sure, definitely no). For this section, provide reference to official recommendations or published literature which used this method, or explain better your decision and way of reasoning.

How is the score for drivers calculated? There is a part about this in the Methods, but I am still unsure what the number 1.43 represents and what would a higher score mean that IS a driver… Usually, something is considered a driver if there are significant differences in the group of questions (dimension) among the willing and hesitant. So please provide a more detailed explanation.

Results

Line 186: Please convert RMB currency into USD for better understanding and comparison. You might even add (in the Discussion) some context, such as the average income in urban or rural areas, or in selected districts.

I am wondering if there were any interesting results/differences if the groups were divided in a different way, by comparing hesitant and refusal group. This might be interesting, even to know there were no significant differences, but also from the point of behavior change – to see what would take to move someone from the outright refusal to the “not sure” group.

Figure 2 – Please find a way to add the average score next to each item so it is easier to read and compare.

Discussion

Considering the importance of cost of vaccination in the reasoning and decision of the participants, it would be useful to understand the context around this a bit better – did all the patients know the cost when they were asked about willingness, and could they take the cost into account. Most countries offer free influenca vaccination to vulnerable groups, so this might be important for comparison.

Can you explain the “pay-it-forward” model for vaccination?

In Limitations: you state many more limitations than what I see – you are more critical to your work than maybe necessary. Why do you think your group of patients (and how) differs from the average patient in China?

Conclusions

In conclusions, please underline the MAIN factors influencing willingness according to your studies and main recommendations in line with those factors.

You mention financial incentives, but I cannot really see from your study results that might be a factor (other than free vaccination). I don’t think this can be a conclusion of your study.

Comments on the Quality of English Language

The English language is very good, except for some terms which need clarification, that I have put in my review report.

Author Response

Dear reviewer,

We really appreciate your favorable and constructive comments that have helped us to improve this manuscript. Meanwhile, we appreciate having the gracious opportunity to revise the manuscript. After careful consideration, we provided point-by-point responses to your comments. And in the revised version of the manuscript, we marked all changes in red so that you can find them quickly. Additionally, we’ve edited the text for language, grammar, and improved clarity. Ambiguous statements have been rephrased for precision, and all terms and statistical data have been checked for accuracy.

If you think the manuscript still needs more revisions, please continue to give us feedback. Thank you very much. Looking forward to your reply.

                                                                                                             Best regards,

                                                                                                             Jul 28th, 2024

Reviewer 2

Comments and Suggestions for Authors

The Manuscript, Enhancing influenza vaccination among patients with type 2 diabetes mellitus in Chongqing, China: a cross-sectional analysis using behavioural and social driver tools” represents an attempt to operationalize the Behavior and Social Drivers (BeSD) tools to better understand barriers and motivators to influenza vaccination among patients with type 2 diabetes mellitus and propose methods to increase the vaccination coverage.

I congratulate the Authors on a very interesting study and very good presentation of the results. The topic of the paper is very relevant, the tools used in the research are up-to-date and studies of this type are more than welcome in the field of using behavior science to better understand vaccination behavior and improve it, especially in vulnerable populations. The Authors should address several comments to better explain their work, and some minor corrections, after which the paper would be recommended for publication.

Abstract

The abstract is well written and includes all the primary information necessary to understand the work. Two major comments arise here:

Comment 1: The conclusions seem like they do not follow the Results section. If the main drivers of vaccination (modifiable) were cost/income, complications, and previous vaccination-I suggest the Authors focus on those aspects in the conclusions. Vaccine efficacy and other social drivers are not mentioned with OR and CI, which is a pity – that is useful information. Being a minority, household and unemployed could also be mentioned as social determinants of health

Response 1: Thank you for your valuable feedback. We appreciate your comments and suggestions for improving our manuscript. We have modified the conclusions in the abstract of the revised manuscript (Page 1, Line 36-37, marked in red). As mentioned in the methods, individuals who selected either "not sure" or "definitely not" were required to complete the hesitancy scale (n=683), while those choosing "definitely yes" responded to the acceptance scale (n=989). Unfortunately, this design means that drivers/barriers are not presented with OR and CI in our current analysis. We acknowledge this limitation and will incorporate these factors into our analysis in future studies to provide a more comprehensive understanding of the drivers and barriers to vaccination willingness.

Comment 2: Please explain or rewrite: “household individual”

Response 2: Thank you for pointing out the ambiguity in the term "household individual." We acknowledge that this term may be unclear. In our context, "household individual" refers to individuals within a household who are unemployed. We’ve revised the term "household or unemployed individual" to "unemployed individuals" throughout the manuscript. Specifically, we have replaced "being household or unemployed individuals" with "being unemployed individuals" in the abstract to ensure clarity. (Page 1, Line 30, marked in red)

Comment 3: Methods: if possible, it would be great to understand how vaccination behavior was measured (scale 1-5 for willingness) and how was the BeSD questionnaire constructed (e.g. used as is in the WHO document)

Response 3: Thank you for your insightful suggestions. We appreciate the need for greater clarity on how vaccination behavior was measured and how the BeSD questionnaire was constructed. In our study, vaccination behaviour was measured using a 3-point Likert scale, with answers ranging from 0 = “disagree” to 2 = “agree,” to identify driving factors or barriers that affected influenza vaccination (Page 4, Line 148-151). Regarding the BeSD questionnaire, we utilized the framework provided in the WHO document, the technical guidelines for seasonal influenza vaccination in China (2022–2023), and existing research in the field of influenza vaccination (Page 3, Line 115-117). We’ve clarified these details in the Methods section of the manuscript.

Introduction

Comment 4:  Line 42-not sure all the complications are “post-infection”, they are just “complications of infection”

Response 4: Thank you for your suggestions. We agree that the term “post-infection” may be misleading. To enhance clarity and accuracy, we have revised the “post-infection” to “complications” in the Introduction section of the manuscript. (Page 2, Line 45, marked in red)

Comment 5:  Please end the Introduction with a clear AIM of the study. E.g. The aim of this study was to identify

Response 5: Thank you for your valuable suggestion. We acknowledge the importance of explicitly stating the aim of the study at the end of the Introduction to provide a clear and focused direction for the readers. We’ve added a clear statement of the study's aim at the end of the Introduction section of the manuscript. (Page 2, Line 77-79, marked in red)

Methods

Comment 6:  Please provide an English version of the full questionnaire as Supplementary material to allow for reproducibility and repeatability of your results.

Response 6: Thank you for your valuable suggestion. We’ve provided the English version of the questionnaire as supplementary material along with the revised manuscript.

Comment 7:  Please provide an explanation why you opted for a 3-point (Likert) scale for the willingness or hesitancy scale and how you decided for the outcome options (definitely yes, not sure, definitely no). For this section, provide reference to official recommendations or published literature which used this method, or explain better your decision and way of reasoning.

Response 7:We appreciate your insightful comment regarding the choice of a 3-point Likert scale for assessing willingness or hesitancy towards influenza vaccination. Our decision to use a 3-point scale for the willingness or hesitancy scale was based on several considerations:

1)Simplicity and clarity: A 3-point scale provides clear and straightforward response options, which can reduce the cognitive burden on respondents, especially in a survey targeting older adults and individuals with chronic health conditions like type 2 diabetes mellitus (T2DM). Simplified scales have been shown to improve response rates and data quality in similar populations 1.

2)Minimizing central tendency bias: By using a 3-point scale, we aim to minimize the central tendency bias often observed with more granular scales (e.g., 5-point or 7-point Likert scales). Respondents may be more likely to choose a neutral midpoint in more extensive scales, which can obscure true willingness or hesitancy2.

3)Evidence from previous researches: Several studies have employed a 3-point scale to assess vaccination intent, finding it effective in capturing respondents' clear positions on the matter. For instance, Dib et al3. used a 3-point scale to evaluate the determinants of human papillomavirus (HPV) vaccine hesitancy in France, reporting reliable and valid results.

   The outcome options "definitely yes," "not sure," and "definitely no" were chosen based on the definition of vaccine hesitancy and to capture the spectrum of vaccination willingness. Vaccine hesitancy is defined as a delay in acceptance or refusal of vaccination despite the availability of vaccination services4. Additionally, the outcome options are consistent with those used in previous studies examining healthcare workers' attitudes towards influenza vaccination5.

Reference:

  • Lewis-Beck, M., Bryman, A., & Futing Liao, T. (Eds.) (2004). The SAGE Encyclopedia of Social Science Research Methods. SAGE Publishing.
  • Revilla, M. A., Saris, W. E., & Krosnick, J. A. (2014). Choosing the Number of Categories in Agree-Disagree Scales. Sociological Methods & Research, 43(1), 73-97.

[3] Dib F, Mayaud P, Launay O, Chauvin P; FSQD-HPVH Study Group. Design and content validation of a survey questionnaire assessing the determinants of human papillomavirus (HPV) vaccine hesitancy in France: A reactive Delphi study. Vaccine. 2020 Sep 3;38(39):6127-6140.

[4] MacDonald NE; SAGE Working Group on Vaccine Hesitancy. Vaccine hesitancy: Definition, scope and determinants. Vaccine. 2015,33(34):4161-4.

[5] Jiang B, Cao Y, Qian J, Jiang M, Huang Q, Sun Y, et al. Healthcare Workers' Attitudes toward Influenza Vaccination: A Behaviour and Social Drivers Survey. Vaccines. 2023;11(1):143.

Comment 8:  How is the score for drivers calculated? There is a part about this in the Methods, but I am still unsure what the number 1.43 represents and what would a higher score mean that IS a driver… Usually, something is considered a driver if there are significant differences in the group of questions (dimension) among the willing and hesitant. So please provide a more detailed explanation.

Response 8: Thank you for your valuable feedback and for pointing out the need for further clarification regarding the calculation of the score for drivers. The score for each driver was calculated based on the responses to the relevant questions using a 3-point Likert scale. Specifically, each response was assigned a value (0 = “disagree,” 1 = “Neutral,” 2 = “agree”), and the scores for each question within a dimension were summed to provide an overall score for that driver. For example, one of the drivers, "Confidence in the safety of IV," had a mean score of 1.43. This score was calculated as follows: 90 participants chose “disagree,” 383 participants chose “neutral,” and 516 participants chose “agree.” The score for this item was therefore calculated as (0×90+1×383+2×516)÷989=1.43. A higher average score indicates a stronger agreement among respondents that the factor is a driver influencing their vaccination behavior. Additionally, each drivers or barriers were designed according to the BeSD framework and the latest research in the field of influenza vaccination1,2.

Reference:

[1]Jiang B, Cao Y, Qian J, Jiang M, Huang Q, Sun Y, et al. Healthcare Workers' Attitudes toward Influenza Vaccination: A Behaviour and Social Drivers Survey. Vaccines. 2023;11(1):143.

[2] Yang L, Nan H, Liang J, Chan YH, Chan L, Sum RW, et al. Influenza vaccination in older people with diabetes and their household contacts. Vaccine. 2017;35(6):889-96.

Results

Comment 9:  Line 186: Please convert RMB currency into USD for better understanding and comparison. You might even add (in the Discussion) some context, such as the average income in urban or rural areas, or in selected districts.

Response 9: Thank you for your insightful comment. We appreciate the suggestion to convert RMB currency into USD for better understanding and comparison. To facilitate this without altering the specific data in the tables, we have included a conversion note below the relevant tables that reads: 1 USD≈7.12 RMB (as of the time of the study). This conversion rate will help readers better understand and compare the costs presented in RMB (Page 6, Line 198-199, marked in red). We’ve also added context regarding the average incomes in urban and rural areas to provide a clearer understanding of the economic impact of vaccination costs in the Discussion section of the manuscript. (Page 13, Line 363-367, marked in red).

Comment 10:  I am wondering if there were any interesting results/differences if the groups were divided in a different way, by comparing hesitant and refusal group. This might be interesting, even to know there were no significant differences, but also from the point of behavior change – to see what would take to move someone from the outright refusal to the “not sure” group.

Response 10: Thank you for your insightful suggestion. According to the definition of vaccine hesitancy, which refers to the delay in acceptance or refusal of vaccines despite the availability of vaccination services, the hesitancy group includes both individuals who chose “not sure” and those who chose “definitely not.” In this study, we did not further subdivide the hesitant group. However, we acknowledge the importance of understanding the nuances between these subgroups. Our future research will aim to compare the hesitant and refusal groups to further identify any significant differences in their vaccination intentions.

Comment 11:  Figure 2–Please find a way to add the average score next to each item so it is easier to read and compare.

Response 11: Thank you for your insightful comment. We’ve added the average score next to each item in Figure 2 and Figure 3 in the manuscript to enhance readability and comparability.

Discussion

Comment 12:  Considering the importance of cost of vaccination in the reasoning and decision of the participants, it would be useful to understand the context around this a bit better – did all the patients know the cost when they were asked about willingness, and could they take the cost into account. Most countries offer free influenza vaccination to vulnerable groups, so this might be important for comparison.

Response 12: Thank you for your insightful comment regarding the importance of understanding the context around the cost of vaccination. We acknowledge that the cost of vaccination plays a crucial role in the decision-making process of participants. In our study, the participants' awareness of the cost of vaccination was indeed an important factor. However, due to the limitations of our data collection methods, we could not ensure that every participant was fully aware of the exact cost when they were asked about their willingness to get vaccinated. Future research will aim to address this gap by providing detailed cost information to participants to better understand its impact on their vaccination decisions. We agree that comparing our findings with countries where influenza vaccination is offered free of charge to vulnerable groups is important. This comparison can provide valuable insights into how financial barriers affect vaccination rates and can guide the development of policies to improve vaccine accessibility. Currently, Chongqing does not have policies for free influenza vaccination or insurance coverage for vaccination costs. Therefore, we have referred to the experiences of other countries or regions in our discussion to provide a basis for improving influenza vaccination coverage rates.

Comment 13:  Can you explain the “pay-it-forward” model for vaccination?

Response 13:Thank you for your valuable comment. The "pay-it-forward" model is an innovative approach to increase vaccine uptake by leveraging social altruism and community support. In this model, individuals receive free vaccination with the understanding that they will then contribute to a fund that helps pay for the next person's vaccination. This creates a chain of goodwill, where each vaccinated individual becomes a part of the effort to ensure others can also be vaccinated. We’ve explained the “pay-it-forward’’ model (a free influenza vaccine and an opportunity to donate financially to support vaccination of other individuals) in the discussion section of the manuscript (Page 13, Line 358-359, marked in red)

Reference: Wu D, Jin C, Bessame K, et al. Effectiveness of a pay-it-forward intervention compared with user-paid vaccination to improve influenza vaccine uptake and community engagement among children and older adults in China: a quasi-experimental pragmatic trial. Lancet Infect Dis. 2022 Oct;22(10):1484-1492. doi: 10.1016/S1473-3099(22)00346-2.

Comment 14:  In Limitations: you state many more limitations than what I see – you are more critical to your work than maybe necessary. Why do you think your group of patients (and how) differs from the average patient in China?

Response 14: Thank you for your valuable feedback. We appreciate your observation regarding the limitations stated in our manuscript. We recognize that our initial approach may have been overly critical. However, we believe that a thorough and detailed discussion of limitations contributes to the scientific rigor of our study. Our group of patients may differ from the average patient in China due to specific regional socioeconomic factors, access to healthcare, and levels of health literacy unique to our study population. Despite these differences, our findings still provide meaningful insights into vaccination behaviors and can contribute to the development of targeted public health strategies.

Conclusions

Comment 15: In conclusions, please underline the MAIN factors influencing willingness according to your studies and main recommendations in line with those factors.

Response 15:Thank you for your insightful comments on our manuscript. Based on your suggestion, we’ve underlined the main factors influencing the willingness to receive influenza vaccination among patients with T2DM in Chongqing, China, and have provided recommendations in line with these factors. We’ve rewrite conclusions section of the manuscript (Page 13, Line 388-399, marked in red)

Comment 16:  You mention financial incentives, but I cannot really see from your study results that might be a factor (other than free vaccination). I don’t think this can be a conclusion of your study.

Response 16Thank you for your valuable feedback. We understand your concern regarding the inclusion of financial incentives in our conclusions. Our study primarily highlighted the significance of free vaccination, higher monthly household income per capita (>5000 RMB)had higher vaccination willingness, and the BeSD survey indicated that high vaccine cost was a substantial barrier to vaccination willingness, scoring significantly higher than other barriers. Additionally, our survey demonstrated that providing free influenza vaccination was a compelling incentive. Therefore, we suggest that addressing cost barriers, including exploring financial incentives, could be important for future strategies to enhance vaccine uptake. We’ve rewrite conclusions section of the manuscript (Page 13, Line 388-399, marked in red)

Reviewer 3 Report

Comments and Suggestions for Authors

Manuscript ID: Vaccines-3108266

Title: Enhancing Influenza Vaccination among Patients with Type 2 Diabetes Mellitus in Chongqing, China: A Cross-Sectional Analysis using Behavioural and Social Driver Tools

COMMENTS TO THE AUTHORS

General comments

The authors investigated willingness of type 2 diabetes mellitus (T2DM) patients in Chongqing, China, to receive the influenza vaccination during the 2023/2024 season, using behavioral and social drivers (BeSD) tools. This is important for improving vaccination uptake in high-risk groups. However, more details and clarifications need to be added to the manuscript. Please see below examples of changes that may make this manuscript easier to read and more useful for its intended purpose.

Specific comments for revision:

a)      Major

    • Please add details on how the logistic regression was performed: how variables were selected to be included in the model, model fit etc…
    • Line 88: some of the exclusion criteria are redundant. Please remove from exclusion criteria: 1) diagnosed with other types of DM, 2) local residence for less than 6 months, 3) individuals under the age of 18 years.
    • Line 119: The authors stated “average monthly household income per capita”. Aren’t these individual household data? Not clear what this variable means. More details about this variable or any other potential area variable may need to be added.
    • Line 182: only one measure of dispersion suffices. Please keep SD and remove the range. Also, please add SD to mean duration of T2DM.
    • Table 3: some priority groups included in this table don’t make sense to me like “3) individuals with DM” and “9) Children 6–59 months of age” are confusing. Don’t all participants have diabetes and are much older? The title says N = 1,672. Please clarify.

b)      Minor

    • Abstract, line 27: typo in “Odds”. Also delete extra comma.
    • Manuscript: please add space in “95%CI”.
    • Please replace “DM” with diabetes throughout the manuscript.
    • Figure 1: insert a space in “Fivedistricts”. Delete Obvious”.
    • May be better to remove figures 2 and 3, or move them to a technical supplement for better flow of the manuscript.

Author Response

Dear reviewer,

We really appreciate your favorable and constructive comments that have helped us to improve this manuscript. Meanwhile, we appreciate having the gracious opportunity to revise the manuscript. After careful consideration, we provided point-by-point responses to your comments. And in the revised version of the manuscript, we marked all changes in red so that you can find them quickly. Additionally, we’ve edited the text for language, grammar, and improved clarity. Ambiguous statements have been rephrased for precision, and all terms and statistical data have been checked for accuracy.

If you think the manuscript still needs more revisions, please continue to give us feedback. Thank you very much. Looking forward to your reply.

                                                                                                     Best regards,

                                                                                                     Jul 28th, 2024

Reviewer 3

COMMENTS TO THE AUTHORS

General comments

The authors investigated willingness of type 2 diabetes mellitus (T2DM) patients in Chongqing, China, to receive the influenza vaccination during the 2023/2024 season, using behavioral and social drivers (BeSD) tools. This is important for improving vaccination uptake in high-risk groups. However, more details and clarifications need to be added to the manuscript. Please see below examples of changes that may make this manuscript easier to read and more useful for its intended purpose.

Specific comments for revision:

  1. a) Major

Comment 1: Please add details on how the logistic regression was performed: how variables were selected to be included in the model, model fit etc…

Response 1: Thank you for your valuable feedback and insightful comments on our manuscript. We appreciate your suggestions for improving the clarity and detail of our methodology section, particularly regarding the logistic regression analysis. We’ve modified in the method section(2.4 Statistical Analyses) in the revised manuscript (Page 5, Line 177-180, marked in red)

Comment 2: Line 88: some of the exclusion criteria are redundant. Please remove from exclusion criteria: 1) diagnosed with other types of DM, 2) local residence for less than 6 months, 3) individuals under the age of 18 years.

Response 2: Thank you for your insightful comment. We’ve removed the redundant exclusion criteria in the method section of the revised manuscript.(Page 2, Line 90, marked in red)

Comment 3: Line 119: The authors stated “average monthly household income per capita”. Aren’t these individual household data? Not clear what this variable means. More details about this variable or any other potential area variable may need to be added.

Response 3: Thank you for your valuable feedback. The variable "average monthly household income per capita" represents the average monthly income for each member of a household. This measure is calculated by dividing the total household income by the number of household members. The inclusion of this variable is designed to assess the potential influence of personal income on vaccination willingness. We have provided additional details on this variable in the methods section of the revised manuscript (Page 3, Line 122-123, marked in red).

Comment 4: Line 182: only one measure of dispersion suffices. Please keep SD and remove the range. Also, please add SD to mean duration of T2DM.

Response 4: Thank you for your suggestion. We’ve keep SD and remove the range of age, and added SD to mean duration of T2DM in the results section of the revised manuscript.(Page 5, Line 189, marked in red)

Comment 5: Table 3: some priority groups included in this table don’t make sense to me like “3) individuals with DM” and “9) Children 6–59 months of age” are confusing. Don’t all participants have diabetes and are much older? The title says N = 1,672. Please clarify.

Response 5: The entry for priority groups for influenza vaccination in Table 3 refers to the technical guidelines for seasonal influenza vaccination in China (2022-2023) to further explore the knowledge rate of diabetic patients to themselves and other priority groups for influenza vaccination, which will help us identify and target those with low awareness for more effective vaccination campaigns. Additionally, we’ve provide an English version of the full questionnaire as supplementary material to better clarify our results.

  1. b) Minor

Comment 6:Abstract, line 27: typo in “Odds”. Also delete extra comma.

Response 6: Thank you for your valuable feedback. We’ve modified in the revised manuscript.(Page 1, Line 29, marked in red)

Comment 7:Manuscript: please add space in “95%CI”.

Response 7: Thank you for your suggestions. We’ve added space in “95% CI” throughout the revised manuscript.

Comment 8:Please replace “DM” with diabetes throughout the manuscript.

Response 8: Thank you for your valuable feedback. We’ve replaced “DM” with diabetes throughout in the revised manuscript. 

Comment 9:Figure 1: insert a space in “Fivedistricts”. Delete Obvious”.

Response 9: Thank you for your valuable feedback. We’ve inserted a space in “Five districts”. and delete Obvious in the Figure 1 in the revised manuscript. 

Comment 10:May be better to remove figures 2 and 3, or move them to a technical supplement for better flow of the manuscript.

Response 10: Thank you for your insightful suggestions. We acknowledge the importance of Figures 2 and 3, which present the behavioral and social drivers (BeSD) survey results, in exploring the drivers and barriers influencing influenza vaccination willingness. These figures are key outcomes of our study and thus have been retained in the revised manuscript. Additionally, we’ve enhanced these figures by adding the average score next to each item, facilitating easier reading and comparison.

Round 2

Reviewer 3 Report

Comments and Suggestions for Authors

The authors have addressed my comments/concerns except the one related to model building and using 0.05 as the significance level when selecting variables that would go to the multivariate model. Some variables not statistically significant by themselves in univariate analyses may become statistically significant when entered with others in a multivariate model (and vice versa). The authors should use a higher cut-off, maybe 0.20 instead of 0.05 significance level in univariate analysis.

Author Response

Dear reviewer,

Thank you for your positive and constructive feedback on our manuscript. We are grateful for the opportunity to revise our work and have carefully considered your comments. In this round of revision, we focused on refining the text for language and clarity, ensuring that all terms and statistical data were accurately represented. Ambiguous statements were rephrased to enhance precision, and the overall readability of the manuscript was improved. We believe that these changes have further strengthened the manuscript. If you think the manuscript still needs more revisions, please continue to give us feedback. Thank you for your guidance and support. Looking forward to your reply.

                                                                                                             Best regards,

                                                                                                             Jul 31th, 2024

Reviewer 3

Comment:The authors have addressed my comments/concerns except the one related to model building and using 0.05 as the significance level when selecting variables that would go to the multivariate model. Some variables not statistically significant by themselves in univariate analyses may become statistically significant when entered with others in a multivariate model (and vice versa). The authors should use a higher cut-off, maybe 0.20 instead of 0.05 significance level in univariate analysis.

Response:Thank you for your constructive feedback. We appreciate your suggestion regarding the use of a higher cut-off level (0.20) for variable selection in the univariate analysis. However, considering the large sample size of our study, we opted for a more stringent P-value threshold of 0.05 to minimize the risk of including spurious associations. This approach aligns with the "strict inclusion criteria" philosophy, ensuring that only the most robust variables are carried forward into the multivariate model.

   Additionally, we conducted exploratory analyses on variables using a higher cut-off level of 0.20, and found that their inclusion did not alter the primary results. Specifically, in our initial analysis, the variable "Other chronic diseases" had a P-value of exactly 0.05, which we subsequently included in the multivariate regression. Importantly, no variables in the study had P-values falling between 0.05 and 0.20, reinforcing our decision that the choice of 0.05 as a threshold did not impact the overall findings.

   We acknowledge the merit of using a more lenient P-value threshold in certain contexts and will consider this approach in future studies where appropriate. Thank you for highlighting this important aspect of model building.

References:

  • Bursac, Z., Gauss, C.H., Williams, D.K. et al. Purposeful selection of variables in logistic regression. Source Code Biol Med 3, 17 (2008). https://doi.org/10.1186/1751-0473-3-17.
  • Mickey RM, Greenland S. The impact of confounder selection criteria on effect estimation. Am J Epidemiol. 1989;129(1):125-37. doi: 10.1093/oxfordjournals.aje.a115101.
